# Serum high sensitivity C-reactive protein poorly predicts bone mineral density: A NHANES 2017–2020 analysis

**Sarah E. Little-Letsinger** [ORCID] *

Department of Evolutionary Anthropology, Duke University, Durham, North Carolina, United States of America

* sarah.little11@gmail.com

**Data Availability Statement:** All data files are publicly available on the CDC website listed under the NHANES 2017-2020 Pre-pandemic Dataset (https://wwwn.cdc.gov/nchs/nhanes/

## Abstract

A reliable, widely available method to detect osteoporosis prior to fracture is needed. Serum levels of C-reactive protein may independently predict low bone mineral density (BMD) and high fracture risk. Existing empirical data focus on sexually and/or racially homogenous populations. This study tests the hypotheses that: C-reactive protein (1) negatively correlates with BMD and (2) fracture history, and (3) independently predicts BMD and fracture history in a diverse population. NHANES 2017–2020 pre-pandemic cycle data were analyzed in R studio. Strength and direction of relationships (-1 to +1) between variables were determined using Kendall's rank correlation coefficient ($\tau$). Linear models were optimized to predict femoral neck or lumbar spine BMD. C-reactive protein positively correlated with femoral ($\tau$ = 0.09, p<0.0001) and spine BMD ($\tau$ = 0.10, p<0.0001). Individuals identifying as female demonstrated more robust, but still weak, correlations between C-reactive protein and femoral neck ($\tau$ = 0.15, p<0.0001; male, $\tau$ = 0.06, p = 0.051) and spine BMD ($\tau$ = 0.16, p<0.0001; male, $\tau$ = 0.06, p = 0.04). C-reactive protein positively correlated with fracture history ($\tau$ = 0.083, p = 0.0009). C-reactive protein significantly predicted femoral neck ($R^2$ = 0.022, p = 0.0001) and spine BMD ($R^2$ = 0.028, p<0.0001) and fracture history ($R^2$ = 0.015, p<0.0001). Exploratory analyses identified weight was the single best predictor for femoral neck ($R^2$ = 0.24, p<0.0001) and spine BMD ($R^2$ = 0.21, p<0.0001). In sum, C-reactive protein statistically correlates with and predicts femoral neck and spine BMD, but the magnitude is too low to be biologically meaningful. While weight is a more robust predictor, individuals who are overweight or obese account for nearly half of all osteoporotic fractures, limiting the predictive power of this variable at identifying individuals at risk for osteoporosis. Identification of a robust predictor of fracture risk in a diverse population and across of range of body weights and compositions is needed.

## Introduction

Osteoporosis is a bone disorder characterized by low bone mass and mineral density and impaired bone microarchitecture. Osteoporosis carries a significant economic burden, with annual costs expected to exceed 25 billion dollars by 2025 in the United States alone [1–3].

**Funding:** The author received no specific funding for this work.

**Competing interests:** The author has declared that no competing interests exist.

Approximately half of women and one quarter of men over the age of 50 will experience at least one osteoporotic fracture in their lifetime [1]. Osteoporotic fracture is associated with substantial mortality and morbidity. For example, within one year following an osteoporotic hip fracture, approximately one quarter of patients are deceased and over half never regain full function or independence [1]. Osteoporosis is a silent disease, meaning it is typically not detected before the first fracture. Thus, there is a critical need to detect osteoporosis prior to fracture.

Dual-energy x-ray absorptiometry (DXA) scanning to quantify bone mineral density is the gold standard diagnostic tool for osteoporosis. Established diagnostic criteria define osteopenia as a T-score between -1 and -2.5 and osteoporosis as a T-score of -2.5 or lower [4]. However, currently DXA scanning is minimally covered by medical insurance. In fact, organizations like the American Society for Bone and Mineral Research advocate at the congressional level to improve medical coverage of DXA scanning as a means to improve early detection. In tandem with these efforts, there is an apparent need to develop and/or identify a reliable and widely available method to detect osteoporosis risk earlier.

Serum high-sensitivity C-reactive protein (hsCRP), a marker of immune activation, is widely viewed as a significant and independent predictor of low bone mineral density and fracture risk. The highest tertile of hsCRP has been linked to low BMD, elevated bone resorption, bone loss, and increased fracture risk [5–9]. In a meta-analysis on studies investigating the relationship of hsCRP and fracture risk, Mun *et al.* report risk ratios of 1.54–1.57 for individuals with the highest tertile of hsCRP [10]. Mun and colleague's conclusion supports the general view of hsCRP as a predictor of fracture risk. Population-level analysis indicates higher incidence of osteoporotic fracture in non-Hispanic white women [11]. However, studies investigating the relationship between hsCRP and fracture risk/incidence predominantly use racially (i.e., Korean, Caucasian) and/or sexually homogenous populations. Further, studies to date have focused on correlations and hazard or relative risk ratios, rather than predictive modeling, calling into question the physiological utility of statistical associations with hsCRP. So, the question remains: can hsCRP levels predict fracture risk, assessed via BMD and/or history of fracture?

Here, I utilize the NHANES 2017–2020 Pre-Pandemic cycle data, which includes a racially and sexually diverse population, to test associations and predictive power between hsCRP and several metrics relating to osteoporosis. I hypothesize that: (1) HSCRP negatively correlates with femoral neck and lumbar spine BMD and (2) history of fracture, and (3) HSCRP is an independent predictor of BMD and fracture in a diverse population. The specific objectives of this analysis were to determine if, and how, the sexual and racial diversity of the sample population impacts the correlative and predictive relationships between CRP and bone mineral density and fracture risk.

## Materials and methods

Datasets from the NHANES 2017–2020 Pre-Pandemic cycle were downloaded from the CDC website. Participants were sampled from 2017 to March 2020. Downloaded datasets include: P_BODMEAS (n = 10000), P_DEMO (n = 10000), P_DXXFEM (n = 4593), P_DXXSPN (n = 4593), P_HSCRP (n = 10000), P_OSQ ((n = 4987), and P_PAQ (n = 9693). Data were uploaded to R Studio (v2022.07.2+576) and individually processed. Data coded as unanswered, incomplete, and/or invalid were excluded [# excluded per dataset: P_BODMEAS (n = 1370), P_DEMO (n = 452), P_DXXFEM (n = 1048), P_DXXSPN (n = 2472), P_HSCRP (n = 2010), P_OSQ ((n = 419), and P_PAQ (n = 91)]. Per NHANES Analysis Standard Procedures, individuals aged 80+ were grouped and coded as age 80; these individuals were excluded (n = 77).

Data were then pooled into one spreadsheet by respondent number and analyzed (n = 954); only individuals with data in all seven datasets with complete and valid data were included. Code used in data processing, analysis, and graphing can be found in the S1 Appendix.

## Statistical analysis

Kendall's rank correlation coefficient ($\tau$), a non-parametric version of Pearson's correlation, was used to assess the strength and direction of correlative relationships (-1 to +1) between variables. Generalized linear models were optimized to predict femoral neck (FN) or lumbar spine (SP) BMD and fracture history. For statistical comparison by sex, non-parametric, unpaired, two-tailed t-tests were used. For comparison by race, non-parametric one-way ANOVA was used, followed by Wilcoxon signed rank test with adjustment for false discovery rate. Code used for statistical analysis can be found in the S1 Appendix.

## NHANES data collection methods

**Demographics.** All demographic data, including age, sex, race, etc., were collected by self-reporting. Individuals were interviewed in their preferred language and/or through an interpreter if requested. Age was calculated using reported date of birth. Sex was self-selected from a provided list: 1) male or 2) female. Race was self-selected from a provided list: 1) Mexican American or Hispanic, 2) non-Hispanic white, 3) non-Hispanic black, 4) non-Hispanic Asian, or 5) non-Hispanic multiracial.

**Body measures.** Body weight and height were collected were collected by trained staff. Body mass index was calculated as the weight in kilograms divided by height in meters squared, then rounded to one decimal place.

**Bone mineral density.** Dual-energy x-ray absorptiometry (DXA) is the primary method to evaluate bone mineral density and osteoporosis risk clinically. A Hologic DXA scanner was used for all scans (Hologic, Inc., Bedford, MA). Quality control phantoms were scanned daily to ensure accurate calibration of the DXA scanner. For the femoral neck, the left hip was scanned unless the participant self-reported a prior left hip fracture, replacement, or a surgical pin. Participants were excluded if the right hip could not be scanned. For the lumbar spine (L1-L4), patients were excluded from the spine scan if they self-reported a rod in the spine.

**C-reactive protein.** Serum specimens were collected and analyzed for levels of C-reactive protein (CRP) using a two-reagent immunoturbidimetric system. Briefly, the specimen is first combined with Tris buffer, then latex particles coated with mouse anti-human CRP antibodies are added. When human CRP is present in the specimen, complexes are formed that cause in increase in light scattering proportional to CRP concentration. Light absorbance is read against a standard CRP curve to determine CRP serum levels.

**History of osteoporosis or fracture.** Participants were asked to self-report history, number, and site of prior fracture. Interviews were conducted in the preferred language of the participant or using an interpreter.

**Physical activity.** Time spent participating in moderate or vigorous physical activity or time spent sedentary was self-reported. Participants were asked to estimate how many minutes were spent in each physical activity zone on a typical day. Interviews were conducted in the preferred language of the participant or using an interpreter.

**Ethical approval.** Exemption from review or approval was acquired from the Duke University Campus Institutional Review Board. The National Health and Nutrition Examination Survey team received approval by the Center for Disease Control and National Center for Health Statistics Ethics Review Board and acquired informed consent during collection of data included in this analysis.

## Results

The sample population achieved sexual and racial diversity. Descriptive statistics by sex and race for all variables included in this study can be found in Table 1. FN BMD varied significantly ($p<2.2e^{-16}$; Figs 1A and 2A) between males ($0.81 \pm 0.13$ g/cm$^2$) and females ($0.73 \pm 0.13$ g/cm$^2$). This can be seen in Fig 1A, where the histogram for males demonstrates a right-ward shift. Individuals identifying as non-Hispanic (NH) black ($0.82 \pm 0.14$ g/cm$^2$) demonstrated significantly greater FN BMD (Fig 2B) than individuals identifying as Hispanic ($p = 1.2e^{-4}$; $0.77 \pm 0.12$ g/cm$^2$), NH white ($p = 5.7e^{-10}$; $0.74 \pm 0.14$ g/cm$^2$), or NH Asian ($p = 1.1e^{-11}$; $0.72 \pm 0.11$ g/cm$^2$). Likewise, individuals identifying as multiracial ($0.82 \pm 0.13$ g/cm$^2$) demonstrated greater FN BMD compared to those identifying as Hispanic ($p = 0.037$), NH white ($p = 0.003$), or NH Asian ($p = 2e^{-4}$). No significant differences were found between individuals identifying as NH black and multiracial ($p = 0.85$). Individuals identifying as Hispanic demonstrated significantly greater FN BMD than individuals identifying as NH white ($p = 0.012$; $0.74 \pm 0.14$ g/cm$^2$) or NH Asian ($p = 1.2e^{-4}$; $0.72 \pm 0.11$ g/cm$^2$). Individuals identifying as NH white and NH Asian did not differ significantly ($p = 0.11$).

SPN BMD was differed significantly ($p<2.2e^{-16}$; Figs 1B and 2C) between males ($1.06 \pm 0.16$ g/cm$^2$) and females ($0.95 \pm 0.15$ g/cm$^2$). This can be seen in Fig 1B, where the histogram for males demonstrates a right-ward shift. Individuals identifying as NH black ($1.05 \pm 1.7$ g/cm$^2$) demonstrated significantly greater FN BMD (Fig 2D) than individuals identifying as Hispanic ($p = 8.1e^{-4}$; $0.97 \pm 0.15$ g/cm$^2$), NH white ($p = 0.011$; $1.01 \pm 0.15$ g/cm$^2$), or NH Asian ($p = 1.8e^{-11}$; $0.93 \pm 0.16$ g/cm$^2$). Likewise, individuals identifying as multiracial ($1.09 \pm 0.21$ g/cm$^2$) demonstrated greater FN BMD compared to those identifying as Hispanic ($p = 0.0012$), NH white ($p = 0.049$), or NH Asian ($p = 1.1e^{-4}$). Individuals identifying as NH black and multiracial did not differ significantly ($p = 0.33$).

**Table 1. Respondent demographics.**

| | All | Male | Female | Hispanic | NH white | NH black | NH Asian | Multiracial |
|---|---|---|---|---|---|---|---|---|
| Sex or Race # (%) | | 446 (46.8%) | 508 (53.2%) | 233 (24.4%) | 310 (32.5%) | 233 (24.4%) | 151 (15.8%) | 27 (2.8%) |
| Age (yrs) Mean (IQR) | 61.4 (55–67) | 61.0 (55–66) | 61.6 (55–67) | 60.3 (55–65) | 63.1 (57–69) | 61.4 (56–65) | 59.2 (53–63) | 61.3 (54–69) |
| Weight (kg) Mean (IQR) | 79.0 (65.5–89.8) | 85.7 (71.6–95.7) | 73.1 (61.0–83.2) | 77.7 (67.3–86.2) | 82.5 (69.2–93.3) | 84.4 (69.3–96) | 65.4 (57–71.3) | 78.9 (69.4–88.9) |
| BMI (kg/m$^2$) Mean (IQR) | 28.9 (25.0–32.0) | 28.7 (25.1–31.6) | 28.1 (24.7–32.5) | 29.4 (26–32) | 29.4 (25.1–32.7) | 30.3 (25.4–34.4) | 24.2 (22.5–27.3) | 27.2 (24.7–29.5) |
| FN BMD (g/cm$^2$) Mean (SD) | 0.77 ± 0.14 | 0.81 ± 0.13 | 0.73 ± 0.13 | 0.77 ± 0.12 | 0.74 ± 0.14 | 0.82 ± 0.14 | 0.72 ± 0.11 | 0.82 ± 0.13 |
| SPN BMD (g/cm$^2$) Mean (SD) | 1.0 ± 0.16 | 1.06 ± 0.16 | 0.95 ± 0.15 | 0.97 ± 0.15 | 1.01 ± 0.15 | 1.05 ± 0.17 | 0.93 ± 0.16 | 1.09 ± 0.21 |
| hsCRP (mg/L) Median (SD) | 1.78 ± 3.11 | 1.75 ± 2.85 | 1.84 ± 3.33 | 1.86 ± 2.65 | 2.04 ± 3.14 | 2.19 ± 3.66 | 0.97 ± 2.55 | 1.90 ± 2.97 |
| History of Fx Y/N (%) | 118/835 (12.3%) | 60/385 (13.5%) | 58/450 (11.4%) | 23/210 (9.9%) | 63/247 (20.3%) | 17/216 (7.3%) | 8/142 (5.3%) | 7/20 (25.9%) |
| Hip Fx Y/N (%) | 13/940 (1.4%) | 4/441 (0.9%) | 9/499 (1.8%) | 2/231 (0.9%) | 8/302 (2.6%) | 1/232 (0.4%) | 1/149 (0.7%) | 1/26 (3.7%) |
| Wrist Fx Y/N (%) | 91/861 (9.6%) | 51/393 (11.5%) | 40/468 (7.9%) | 17/216 (7.3%) | 48/261 (15.5%) | 14/219 (6.0%) | 6/144 (4.0%) | 6/21 (22.2%) |
| Spine Fx Y/N (%) | 23/930 (2.4%) | 10/435 (2.2%) | 13/495 (2.6%) | 6/227 (2.6%) | 13/297 (4.2%) | 2/231 (0.9%) | 2/148 (1.3%) | 0/27 (0%) |
| No Fx # (%) | 836 (87.6%) | 386 (86.5%) | 450 (88.6%) | 210 (90.1%) | 247 (79.7%) | 216 (92.7%) | 143 (94.7%) | 20 (74.1%) |
| 1 Fx # (%) | 93 (9.7%) | 46 (10.3%) | 47 (9.3%) | 19 (8.2%) | 46 (14.8%) | 16 (6.9%) | 7 (4.6%) | 5 (18.5%) |
| 2 Fx # (%) | 17 (1.8%) | 11 (2.5%) | 6 (1.2%) | 3 (1.3%) | 11 (3.5%) | 1 (0.4%) | 1 (0.7%) | 1 (3.7%) |
| 3 Fx # (%) | 5 (0.5%) | 1 (0.2%) | 4 (0.8%) | 1 (0.4%) | 3 (0.9%) | | | 1 (3.7%) |
| 4 Fx # (%) | 3 (0.3%) | 2 (0.4%) | 1 (0.2%) | | 3 (0.9%) | | | |

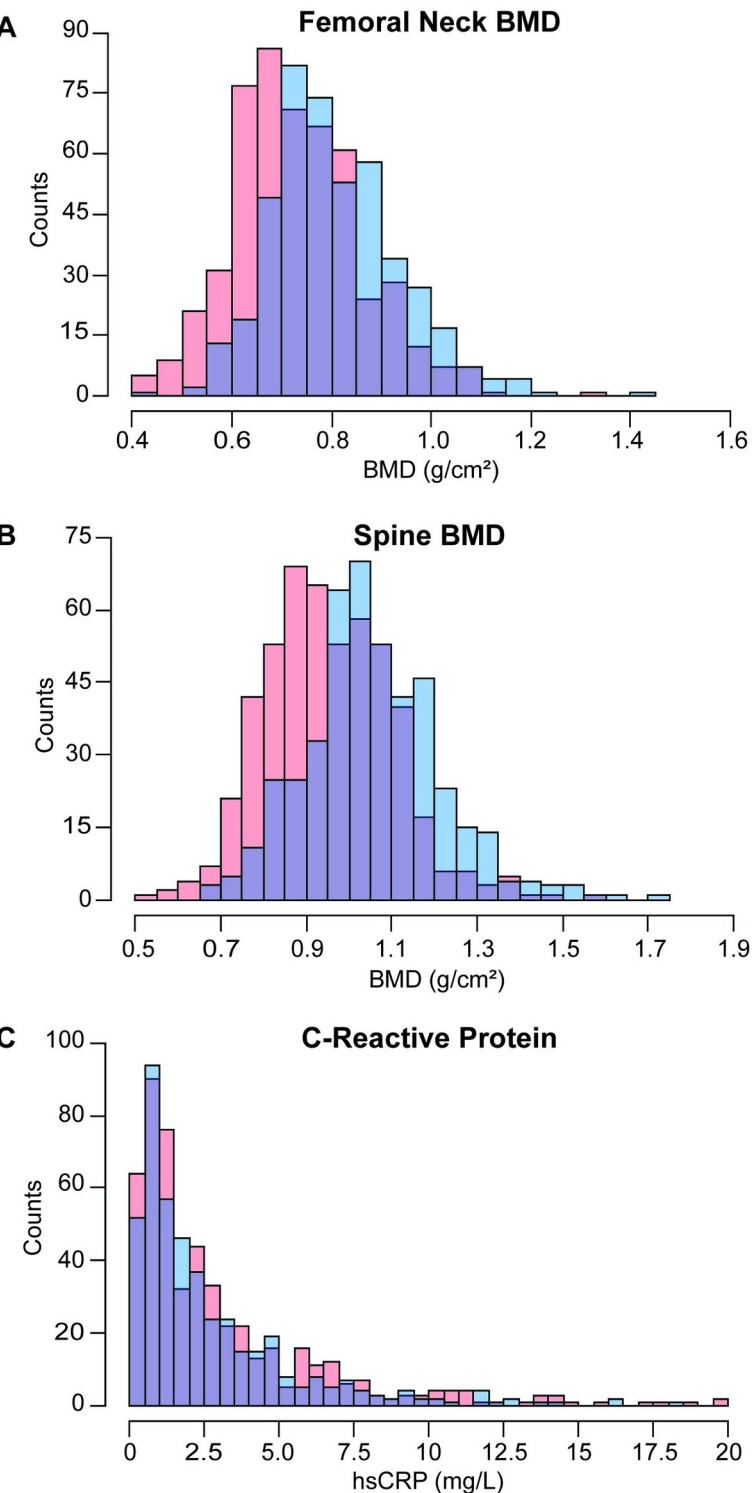

**Fig 1. Femoral neck and spine BMD, but not C-reactive protein, differ by sex.** Distribution of **A)** femoral neck BMD (g/cm²), **B)** spine BMD (g/cm²), and **C)** C-reactive protein (mg/L) in the sample population. Individuals identifying as male are coded in blue, as female in red, and the overlap in purple. Bin size was optimized for each variable using the Freedman-Diaconis method.

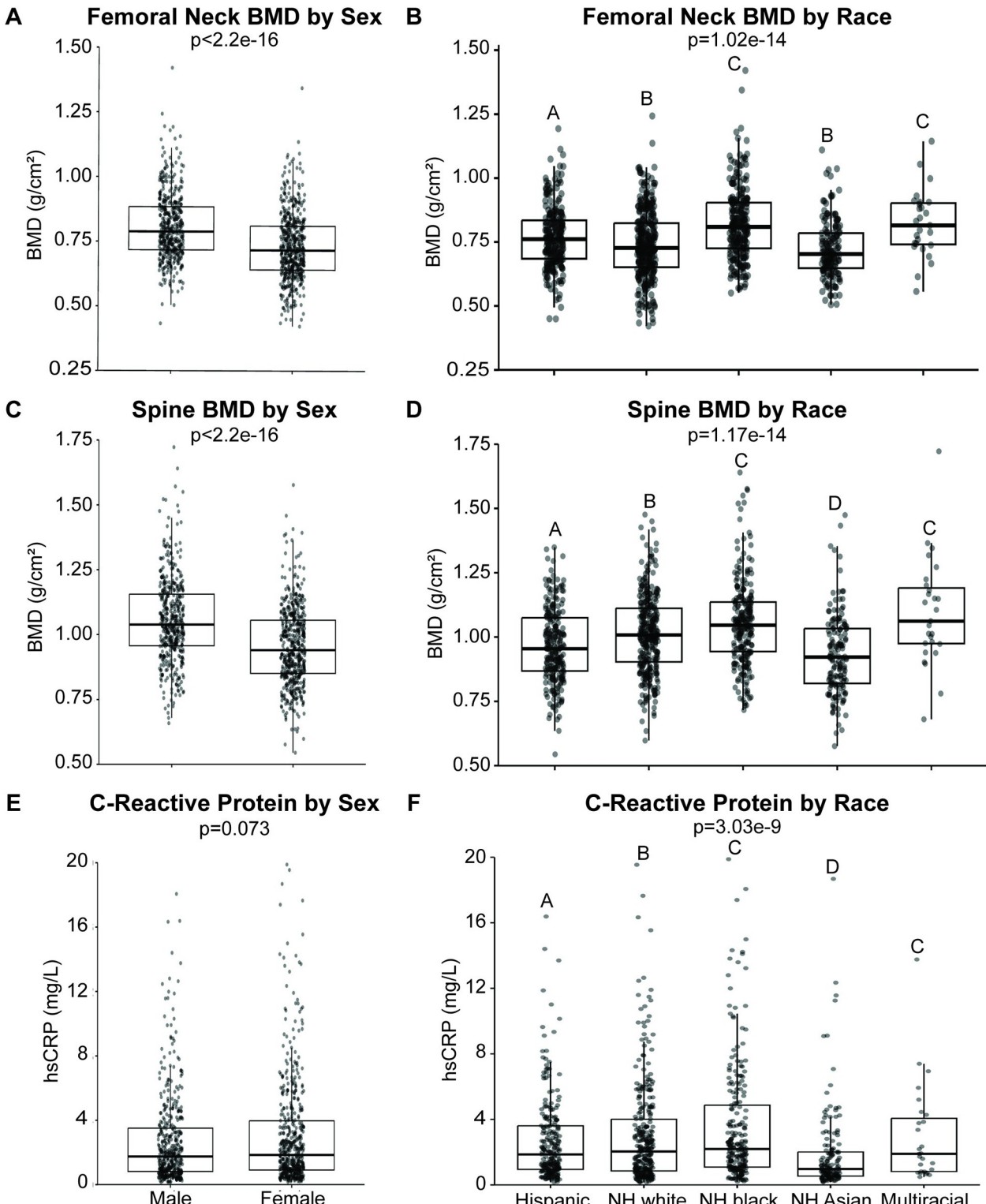

**Fig 2. Variation in BMD and C-reactive protein by sex and race.** Femoral neck BMD (g/cm²) by **A)** sex and **B)** race. Spine BMD (g/cm²) by **C)** sex and **D)** race. C-reactive protein (mg/L) by **E)** sex and **F)** race. Boxplots denote the 1st, 2nd (median), and 3rd quartiles, with all data points plotted. For statistical comparison by sex, non-parametric, unpaired, two-tailed t-tests were used. For comparison by race, non-parametric one-way ANOVA was used, followed by Wilcoxon signed rank test with adjustment for false discovery rate.

hsCRP did not differ significantly (p = 0.073; Figs 1C and 2E) between males (2.7 ± 2.8 g/cm$^2$) and females (3.06 ± 3.32 g/cm$^2$). This is demonstrated in the histogram in Fig 1C, where the male and female histograms largely overlap. Individuals identifying as NH Asian (1.9 ± 2.5 g/cm$^2$) demonstrated significantly lower (Fig 2F) hsCRP compared to those identifying as Hispanic (p = 1.5e$^{-6}$; 2.7 ± 2.7 g/cm$^2$), NH white (p = 1.5e$^{-6}$; 3.0 ± 3.1 g/cm$^2$), NH black (p = 8.8e$^{-10}$; 3.5 ± 3.7 g/cm$^2$), or multiracial (p = 0.013; 3.0 ± 3.0 g/cm$^2$).

hsCRP and FN BMD were weakly, positively correlated ($\tau$ = 0.1022, p = 2.41e$^{-6}$; Fig 3A). hsCRP correlates more robustly with FN BMD in females ($\tau$ = 0.1609, p = 6.27e$^{-8}$; Fig 3C), but does not correlate in males ($\tau$ = 0.0551, p = 0.08; Fig 3E). hsCRP does not correlate with FN BMD in individuals identifying as Hispanic ($\tau$ = 0.0489, p = 0.27; Fig 4A), NH Asian ($\tau$ = 0.0344, p = 0.53; Fig 4D), or multiracial ($\tau$ = 0.0456, p = 0.74; Fig 4E). hsCRP is weakly positively correlated with FN BMD in individuals identifying as NH white ($\tau$ = 0.1117, p = 0.0034; Fig 4B) and NH black ($\tau$ = 0.0874, p = 0.0476; Fig 4C).

hsCRP and SPN BMD were weakly, positively correlated ($\tau$ = 0.1037, p = 1.69e$^{-6}$; Fig 3B). hsCRP correlates more robustly with SPN BMD in females ($\tau$ = 0.1709, p = 8.99e$^{-9}$; Fig 3D), but does not correlate in males ($\tau$ = 0.0512, p = 0.11; Fig 3F). hsCRP does not correlate with SPN BMD in individuals identifying as Hispanic ($\tau$ = 0.0496, p = 0.26; Fig 5A), NH black ($\tau$ = 0.0683, p = 0.12; Fig 5C), NH Asian ($\tau$ = 0.0137, p = 0.80; Fig 5D), or multiracial ($\tau$ = -0.2079, p = 0.13; Fig 5E). hsCRP is weakly positively correlated with SPN BMD in individuals identifying as NH white ($\tau$ = 0.1416, p = 2e$^{-4}$; Fig 5B).

History of Frx, ranging from 0–4 prior fractures, and the distribution across sex and race is in Table 1. hsCRP and history of Frx were weakly, positively correlated ($\tau$ = 0.0919, p = 4.52e$^{-4}$; Fig 6A). History of Frx was weakly, negatively correlated with FN ($\tau$ = -0.0526, p = 0.045; Fig 6B) BMD, but not SPN BMD ($\tau$ = -0.0297, p = 0.26; Fig 6C).

Linear modeling was performed on the whole dataset to determine if hsCRP remains a significant predictor of FN and SPN BMD and history of Frx in a racially and sexually diverse population (Table 2). Log transformation of hsCRP, which was non-normally distributed, yielded a more robust model. hsCRP is a very weak predictor of FN BMD ($R^2_{adj}$ = 0.028, p = 1.21e$^{-7}$), SPN BMD ($R^2_{adj}$ = 0.022, p = 2.14e$^{-6}$) and history of Frx ($R^2_{adj}$ = 0.015, p = 1.08e$^{-4}$). Exploratory modeling was performed to identify a more robust independent predictor of FN and SPN BMD and history of Frx using the variables in the dataset (Table 3). The best model was selected by minimizing AIC. Weight was the best predictor of FN BMD ($R^2_{adj}$ = 0.24, p<2.2e$^{-16}$) and SPN BMD ($R^2_{adj}$ = 0.21, p<2.2e$^{-16}$). Weight demonstrates a ~10-fold increase in model accuracy, or goodness-of-fit, as assessed by $R^2_{adj}$, to predict FN and SPN BMD. The best predictor of history of Frx was race (AIC = 1062.3, race p = 0.143); however, the model failed to reach statistical significance indicating is not a strong predictor.

To explore the relationship between weight and hsCRP, FN BMD, and SPN BMD, exploratory correlation analyses were performed. Weight was positively correlated with hsCRP ($\tau$ = 0.2456, p<2.2e$^{-16}$; Fig 7D), FN BMD ($\tau$ = 0.3376, p<2.2e$^{-16}$; Fig 7E), and SPN BMD ($\tau$ = 0.3424, p<2.2e$^{-16}$; Fig 7F). Weight varied significantly (p<2.2e$^{-16}$; Fig 7A and 7B) between males (85.7 ± 19.5 kg) and females (73.1 ± 16.9 kg). This can be seen in Fig 7A, where the histogram for males demonstrates a right-ward shift. It is possible that weight correlates more strongly with FN and SPN BMD than hsCRP because, unlike hsCRP it follows the same distribution pattern as FN and SPN BMD (Fig 1A and 1B). Individuals identifying as Hispanic (77.7 ± 16.4 kg) demonstrated significantly reduced weight (Fig 7C) than individuals identifying as NH white (p = 0.012; 82.5 ± 19.5 kg), NH black (p = 0.002; 84.4 ± 21.0 kg), or NH Asian (p = 4.6e$^{-15}$; 65.4 ± 12.8 kg). Individuals identifying as NH Asian demonstrated significantly reduced weight compared to those identifying as Hispanic (p = 4.6e$^{-15}$), NH white (p<2.2e$^{-16}$), NH black (p<2.2e$^{-16}$), or multiracial (p = 2.6e$^{-5}$; 78.9 ± 15.2 kg).

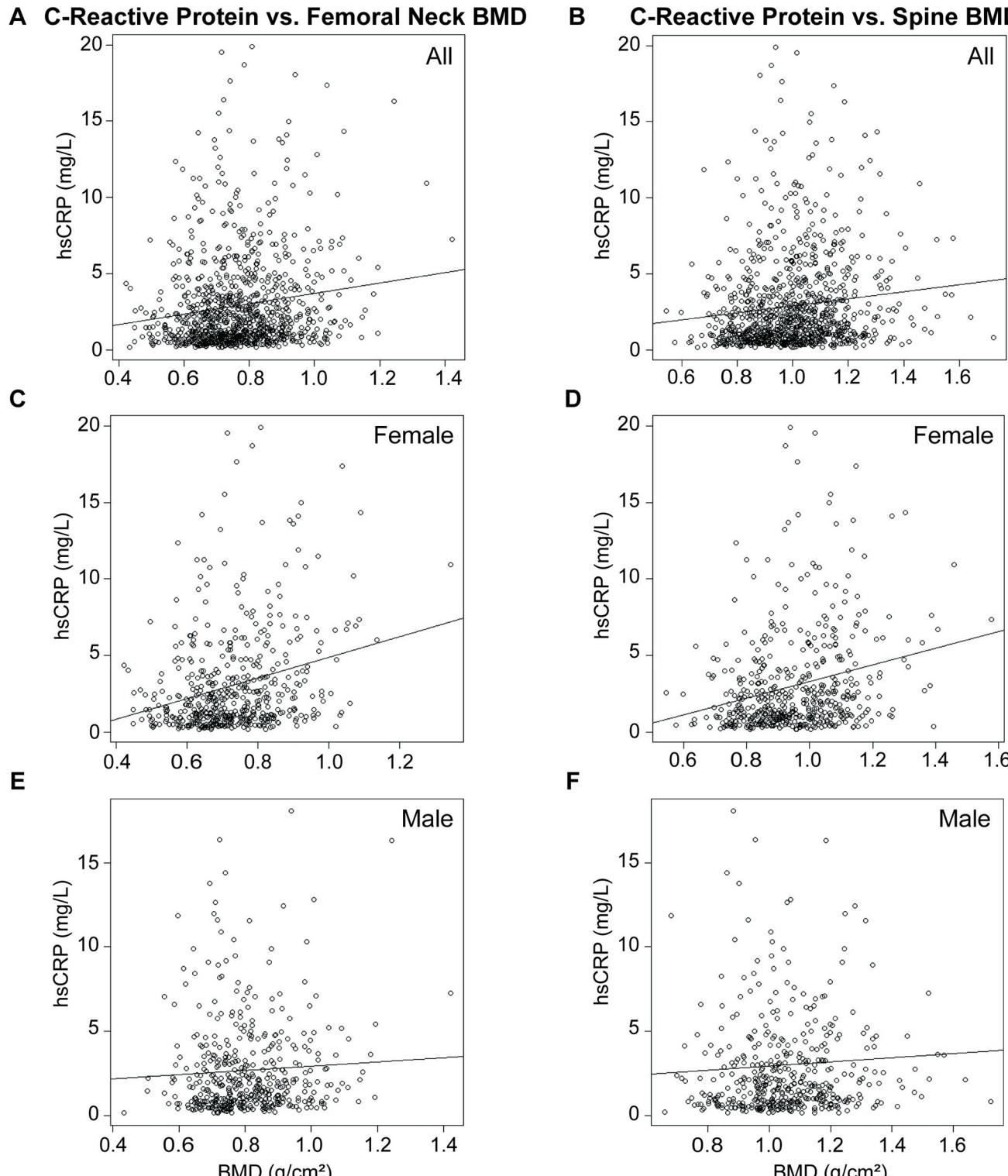

**Fig 3. Correlation between C-reactive protein and BMD by sex.** Correlation between C-reactive protein and femoral neck BMD among **A)** all respondents, **B)** females, and **C)** males. Correlation between C-reactive protein and spine BMD among **D)** all respondents, **E)** females, and **F)** males. Plotted line indicates trendline, with slope of Kendall's tau. Correlations assessed using non-parametric Kendall's rank correlation test.

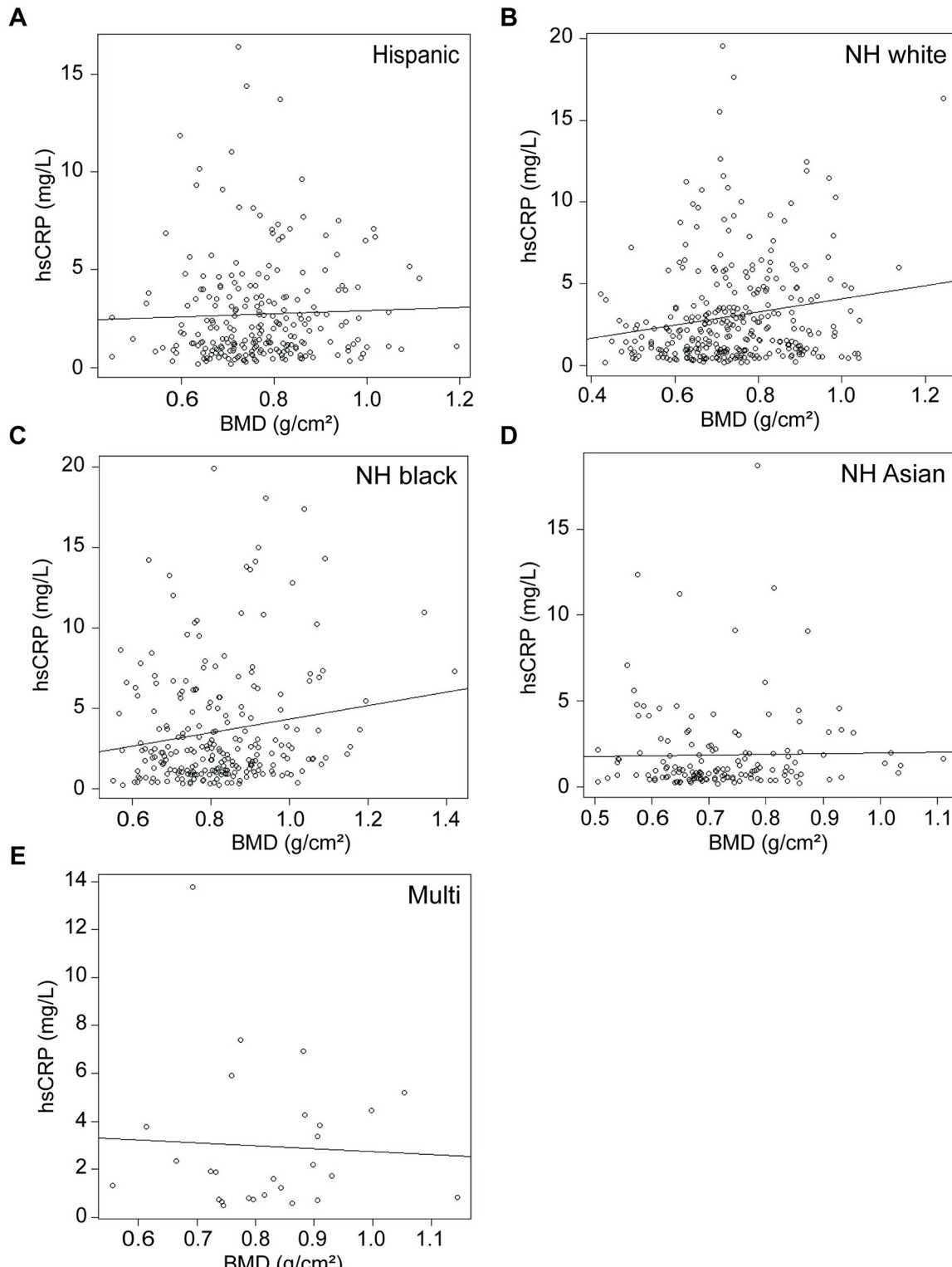

**Fig 4. Correlation between C-reactive protein and femoral neck BMD by race.** Correlation between C-reactive protein and femoral neck BMD among individuals identifying as **A)** Hispanic, **B)** non-Hispanic (NH) white, **C)** NH black, **D)** NH Asian, or **E)** multiracial (multi). Plotted line indicates trendline, with slope of Kendall's tau. Correlations assessed using non-parametric Kendall's rank correlation test.

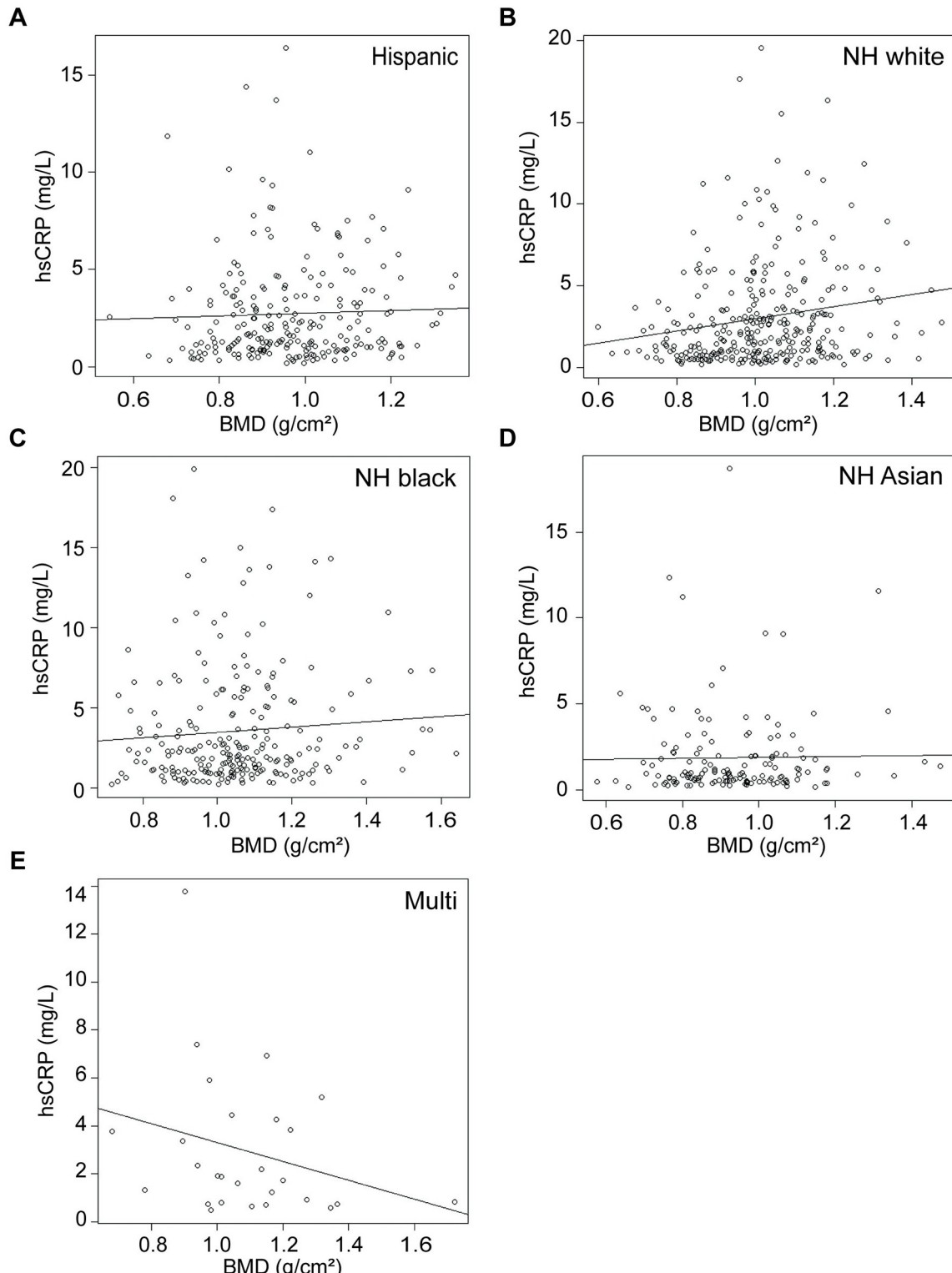

**Fig 5. Correlation between C-reactive protein and spine BMD by race.** Correlation between C-reactive protein and spine BMD among individuals identifying as **A)** Hispanic, **B)** non-Hispanic (NH) white, **C)** NH black, **D)** NH Asian, or **E)** multiracial (multi). Plotted line indicates trendline, with slope of Kendall's tau. Correlations assessed using non-parametric Kendall's rank correlation test.

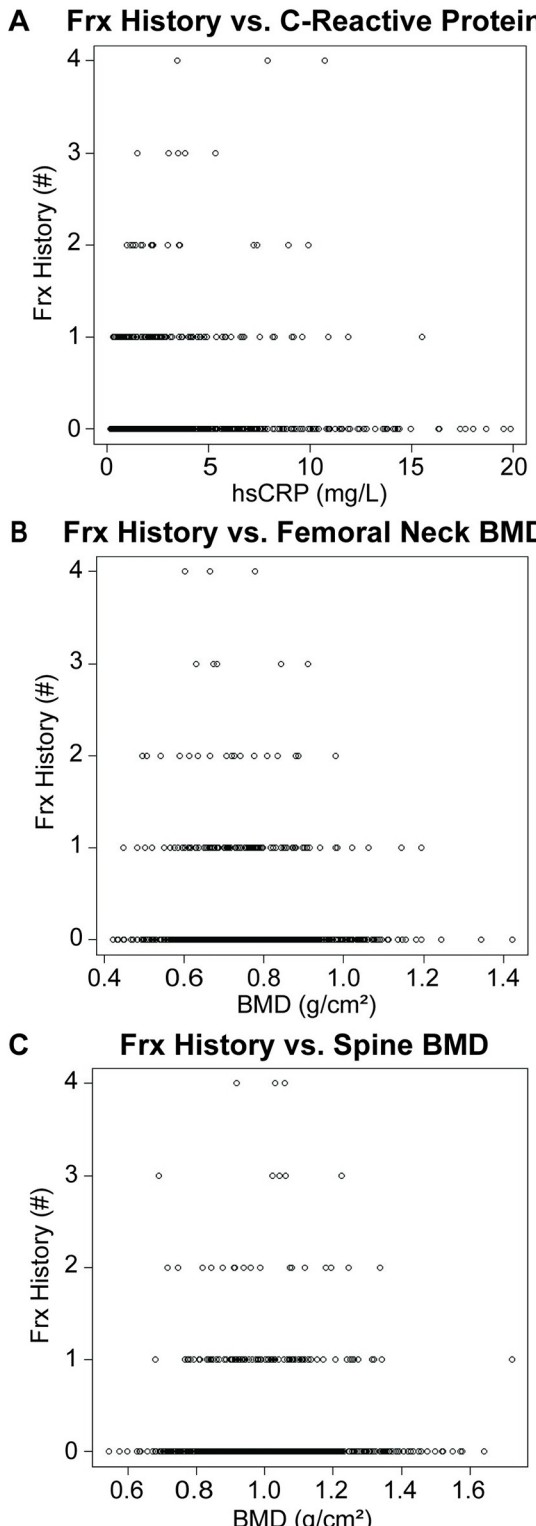

**Fig 6. Correlation between Frx history and C-reactive protein or BMD.** Correlation of Frx history with **A)** C-reactive protein, **B)** femoral neck BMD, and **C)** spine BMD. Correlations assessed using non-parametric Kendall's rank correlation test.

**Table 2. Modeling to predict BMD and history of Frx from C0.**

| Model Outputs | Linear Models with raw hsCRP | | Linear Models with ln(hsCRP) | |
|---|---|---|---|---|
| *FN BMD* | df = 952, RSE = 0.1337, F = 21.5, | | df = 952, RSE = 0.1333, F = 28.4, | |
| | $R^2_{adj}$ = 0.021, p = 3.97e$^{-6}$ | | $R^2_{adj}$ = 0.028, p = 1.21e$^{-7}$ | |
| Intercept | 0.757 | p<1e$^{-16}$ | 0.753 | p<1e$^{-16}$ |
| hsCRP | 0.00645 | p<1e$^{-6}$ | 0.0229 | p<1e$^{-7}$ |
| *SPN BMD* | df = 952, RSE = 0.1662, F = 14.1, | | df = 952, RSE = 0.1615, F = 22.8, | |
| | $R^2_{adj}$ = 0.014, p = 1.88e$^{-4}$ | | $R^2_{adj}$ = 0.022, p = 2.14e$^{-6}$ | |
| Intercept | 0.983 | p<1e$^{-10}$ | 0.987 | p<1e$^{-16}$ |
| hsCRP | 0.00633 | p<0.01 | 0.0249 | p<1e$^{-6}$ |
| *History of Frx* | df = 952, RSE = 0.4891, F = 5.71, | | df = 952, RSE = 0.4868, F = 15.1, | |
| | $R^2_{adj}$ = 0.005, p = 0.017 | | $R^2_{adj}$ = 0.015, p = 1.08e$^{-4}$ | |
| Intercept | 0.126 | p<1e$^{-9}$ | 0.126 | p<1e$^{-12}$ |
| hsCRP | 0.0122 | p<0.05 | 0.0612 | p<1e$^{-4}$ |

Weight and history of Frx are weakly positively correlated ($\tau$ = 0.0857, p = 1.07e$^{-3}$; *not shown*).

## Discussion

Contrary to predictions, hsCRP was weakly and positively associated with BMD at the FN and SPN. hsCRP remained a significant independent predictor of FN and SPN BMD and fracture history in a racially and sexually diverse population; however, the $R^2$ was too low to be biologically meaningful or exert any effect predictive power. Exploratory analyses suggest weight as a better independent predictor of FN and SPN BMD and fracture history, with a 10-fold improvement on $R^2$ and predictive power; however, these metrics remain relatively low, limiting their utility.

Demographic data of FN and SPN BMD were generally as expected. Skeletal sexual dimorphism is well documented, with males maintaining higher BMD at both sites and total-body than females on average. Differences in BMD based on racial identity have, likewise, been previously reported. Individuals identifying as NH white have long been viewed as being at greater risk of osteoporosis due to lower average BMD scores compared to individuals identifying as

**Table 3. Modeling to identify best predictor of BMD.**

| Model Outputs | Model Summary | |
|---|---|---|
| *FN BMD* | df = 952, RSE = 0.1178, F = 302.7, | |
| | $R^2_{adj}$ = 0.24, p<2.2e$^{-16}$ | |
| Intercept | 0.493 | p<1e$^{-16}$ |
| Weight | 0.00346 | p<1e$^{-16}$ |
| *SPN BMD* | df = 952, RSE = 0.145, F = 258.2, | |
| | $R^2_{adj}$ = 0.21, p<2.2e$^{-16}$ | |
| Intercept | 0.691 | p<1e$^{-16}$ |
| Weight | 0.00393 | p<1e$^{-16}$ |
| *History of Frx* | df = 952, AIC = 1062.3, | |
| | Residual Deviance = 1058.3 | |
| Intercept | 1.092 | p<1e$^{-16}$ |
| Race | 0.262 | p = 0.143 |

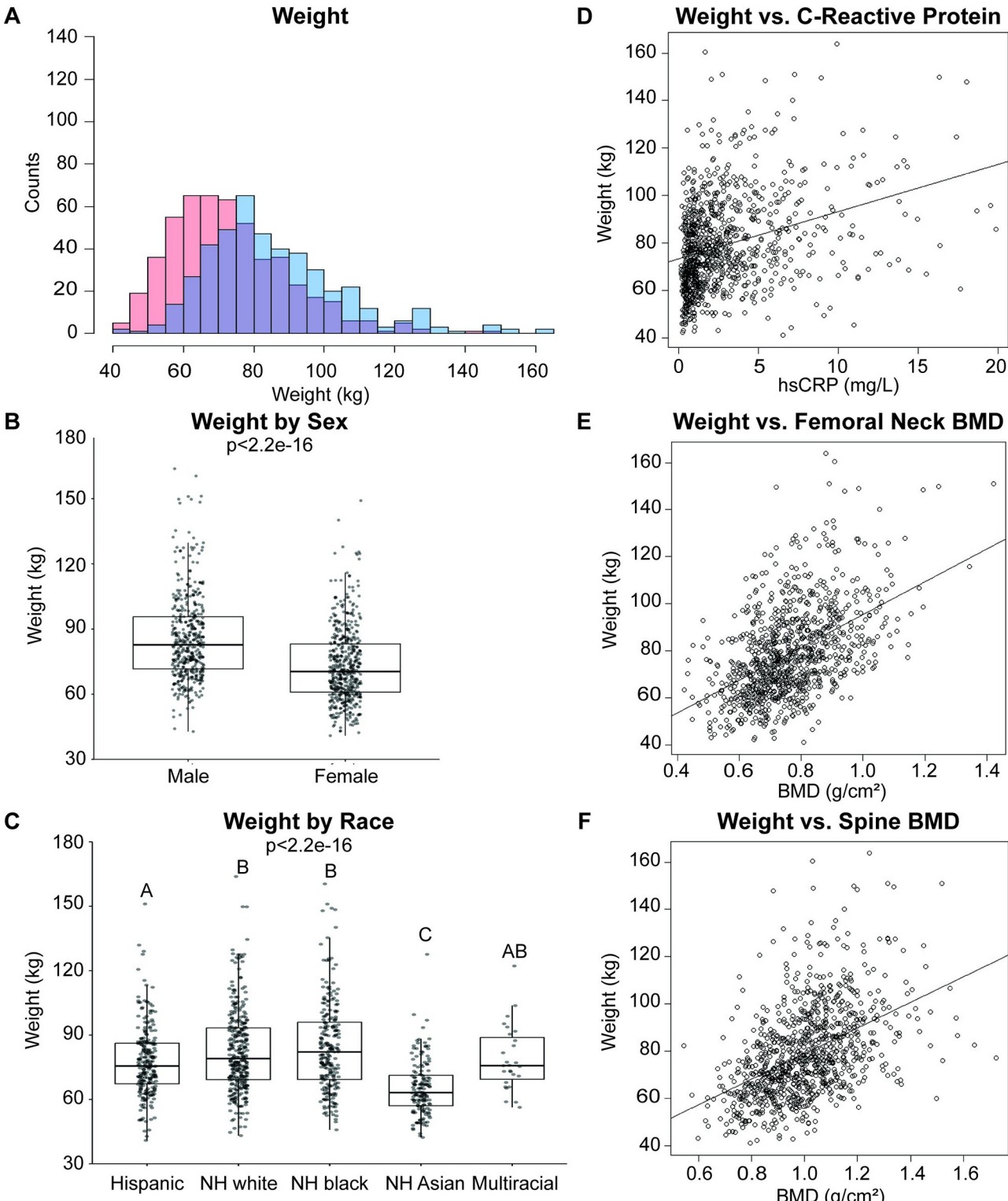

**Fig 7. Weight varies by sex and race, and positively correlates with C-reactive protein and BMD.** Distribution of **A)** weight (kg) in the sample population. Individuals identifying as male are coded in blue, as female in red, and the overlap in purple. Bin size was optimized for each variable using the Freedman-Diaconis method. Weight by **B)** sex and **C)** race. Boxplots denote the 1st, 2nd (median), and 3rd quartiles, with all data points plotted. For statistical comparison by sex, non-parametric, unpaired, two-tailed t-tests were used. For comparison by race, non-parametric one-way ANOVA was used, followed by Wilcoxon signed rank test with adjustment for false discovery rate. Correlation between weight and **D)** C-reactive protein, **E)** femoral

neck BMD, and **F)** spine BMD among all respondents. Plotted line indicates trendline, with slope of Kendall's tau. Correlations assessed using non-parametric Kendall's rank correlation test.

NH black [11, 12]. Some studies have reported individuals identifying as NH Asian to be at a greater or similar risk of osteoporosis as those identifying as NH white, though limited data exist [11, 12]. It is unlikely that such differences reflect genetic differences, as far greater variation exists within than between race. While speculative, it is more likely that these differences are indicative of cultural practices encompassing a range of behavioral and lifestyle factors, including physical activity and diet/nutrition, among others, and may reflect differences in reference weight and stature ranges across racial groups.

hsCRP did not differ by sex or race. In the few studies that compared hsCRP levels across sex in a healthy population demonstrate mixed results, with some concluding no significant differences between sexes [13, 14]; those that do find that females have greater hsCRP levels than males, on average [15, 16]. In a meta-analysis examining associations between race/ethnicity and serum CRP levels, Nazmi & Victora report 14 of the 15 included studies detected significant racial differences in CRP levels [17]. Individuals identifying as NH white demonstrated the lowest CRP levels, while those identifying as Hispanic, NH black, or South Asian had the highest CRP levels [17]. Here, we found individuals identifying as NH Asian demonstrate the lowest mean hsCRP levels relative to those identifying as Hispanic, NH white, NH black, and multiracial. That the testing method is slightly different (*i.e.* CRP versus hsCRP) and the countries sampled included USA, the UK, Finland, Greece, Germany, Canada, Italy, Turkey, and New Zealand rather than the USA alone may be contributing factors. Likewise, the difference in relationships between hsCRP and race found in our sample compared to others may explain the reduced predictive power of hsCRP for BMD. Most studies in the meta-analysis attribute CRP differences to socioeconomic factors [17], which may influence BMD and fracture risk as well.

History of fracture weakly, positively correlated with hsCRP levels and weakly, negatively correlated FN and SPN BMD. Several studies have reported significantly increased relative risk, odds ratios, or hazard ratios for individuals with hsCRP levels in the highest tertile [5, 7–10], suggesting a negative correlation between hsCRP and fracture. Of these studies that evaluated BMD, there was no significant association between BMD and hsCRP, though hsCRP did correlate with serum markers of bone resorption [7]. Ding and colleagues found a significant negative association with BMD and hsCRP, as well as inflammatory makers IL-6 and TNFα [6]; other inflammatory cytokines and circulating markers of immune activation may have greater predictive power than hsCRP and should be evaluated. The demographic pool used in this study had a relatively low proportion of individuals experiencing a fracture (~12.4%) and very low levels of multiple fractures (~2.6%), which may have masked more robust associations.

In contrast to prior published studies on the relationship between hsCRP and BMD or history of fracture, here we show weak, positive correlations between hsCRP and FN BMD, SPN BMD, and history of fracture. One potential explanation for these differences may be our usage of a racially and sexually diverse sample population, especially given known sexual and racial differences in BMD and fracture history. We identify weight as a more robust predictor of FN and SPN BMD, considering the variables identified in Table 1. The relatively small predictor variable pool is a limitation of this work and future research should employ a machine learning approach to more robustly identify predictors and a prediction equation. That weight is the best predictor of FN and SPN BMD is not surprising. However, given that nearly half of

all osteoporotic fractures occur in individuals who are overweight or obese restrains the predictive power of this variable [18]. Indeed, obese and overweight individuals fracture despite high BMD; while the precise biological underpinnings of this are yet to be identified, it may be related to impair bone quality–*e.g.* through increased cortical porosity–which impairs bone strength and leads to fractures not traditionally associated with osteoporosis [19–22]. The complex multivariable etiology of osteoporotic fractures will certainly complicate the search for a single robust predictor. More research is needed to identify a robust predictor of low BMD and high fracture risk, not only in a sexually and racially diverse population, but across of range of body weights and compositions.

In conclusion, serum levels of C-reactive protein statistically correlate with and predict femoral neck and spine BMD in a sexually and racially diverse population. However, as the magnitude is too low to be biologically meaningful, significant caution should be taken in interpreting the clinical significance and application of serum C-reactive protein as a biomarker for osteoporosis. Weight more robustly predicts femoral neck and spine BMD, compared to serum C-reactive protein. Given that overweight and obese individuals account for nearly half of all osteoporotic fractures, the predictive power of weight in identifying individuals at risk for osteoporosis is severely limited. Current research should continue efforts to identify a clinically available biomarker that robustly predicts low bone mineral density and fracture risk in a diverse population and across of range of body weights and compositions. Rates of both osteoporosis and obesity continue to increase, with osteoporosis remaining largely undetected prior to first fracture, lending significant credence to the need for a robust and predictive biomarker.

## Supporting information

**S1 Appendix. Master coding folder.** Code used in data processing, analysis, and graphing. (ZIP)

## Acknowledgments

The author would like to thank Dr. Ayland Letsinger, Dr. Daniel Schmitt, and the Animal Locomotion Lab at Duke University for the much-appreciated conversations, feedback, and support.

## Author Contributions

**Conceptualization:** Sarah E. Little-Letsinger.

**Data curation:** Sarah E. Little-Letsinger.

**Formal analysis:** Sarah E. Little-Letsinger.

**Funding acquisition:** Sarah E. Little-Letsinger.

**Investigation:** Sarah E. Little-Letsinger.

**Methodology:** Sarah E. Little-Letsinger.

**Project administration:** Sarah E. Little-Letsinger.

**Resources:** Sarah E. Little-Letsinger.

**Software:** Sarah E. Little-Letsinger.

**Supervision:** Sarah E. Little-Letsinger.

**Validation:** Sarah E. Little-Letsinger.

**Visualization:** Sarah E. Little-Letsinger.

**Writing – original draft:** Sarah E. Little-Letsinger.

**Writing – review & editing:** Sarah E. Little-Letsinger.

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
