## [Decision Letter · Decision Letter 0]

18 Sep 2023

PONE-D-23-14590Serum High Sensitivity C-Reactive Protein Poorly Predicts Bone Mineral Density: A NHANES 2017-2020 AnalysisPLOS ONE

Dear Dr. Little,

Thank you for submitting your manuscript to PLOS ONE. After careful consideration, we feel that it has merit but does not fully meet PLOS ONE’s publication criteria as it currently stands. Therefore, we invite you to submit a revised version of the manuscript that addresses the points raised during the review process.

We look forward to receiving your revised manuscript.

Kind regards,

Ewa Tomaszewska, DVM Ph.D

Academic Editor

PLOS ONE

Journal Requirements:

"The author would like to thank Dr. Ayland Letsinger, Dr. Daniel Schmitt, and the Animal Locomotion Lab at Duke University for the much-appreciated conversations, feedback, and support. The authors salary is supported by Duke University School of Medicine." 

"The author received no specific funding for this work."

5. We are unable to open your Supporting Information file [NHANESprojectfolder.zip]. Please kindly revise as necessary and re-upload.

Reviewers' comments:

Reviewer's Responses to Questions

**Comments to the Author**

1. Is the manuscript technically sound, and do the data support the conclusions?

Reviewer #1: No

2. Has the statistical analysis been performed appropriately and rigorously? 

Reviewer #1: Yes

3. Have the authors made all data underlying the findings in their manuscript fully available?

Reviewer #1: Yes

4. Is the manuscript presented in an intelligible fashion and written in standard English?

Reviewer #1: Yes

5. Review Comments to the Author

Reviewer #1: I reviewed the manuscript titled “Serum High Sensitivity C-Reactive Protein Poorly Predicts Bone Mineral Density: A NHANES 2017-2020 Analysis” submitted for publication in PLOS One as a research article.

Disclaimers:

-please add keywords.

-it would be appropriate to briefly define the objectives of the analysis in a separate paragraph.

-summarizing the results of the discussion in the form of conclusions in a separate paragraph.

6. PLOS authors have the option to publish the peer review history of their article (what does this mean?). If published, this will include your full peer review and any attached files.

Reviewer #1: **Yes: **Jakub Kosiński

---

## [Author Response · Author response to Decision Letter 0]

20 Sep 2023

Rebuttal Letter: PONE-D-23-14590

Serum high sensitivity C-reactive protein poorly predicts bone mineral density: a NHANES 2017-2020 analysis

I thank the reviewer for their thoughtful review and suggested revisions. Please find below a point-by-point response. I hope you find all comments to be adequately addressed.

Reviewer Comment #1: please add keywords.

• The following keywords have been added to the manuscript on Page 1, Line 19: osteoporosis, fracture risk, diverse population

Reviewer Comment #2: it would be appropriate to briefly define the objectives of the analysis in a separate paragraph.

• The following sentence defining the objectives of the analysis have been added on Page 4, Lines 80-82:

“The specific objectives of this analysis were to determine if, and how, the sexual and racial diversity of the sample population impacts the correlative and predictive relationships between CRP and bone mineral density and fracture risk.”

Reviewer Comment #3: summarizing the results of the discussion in the form of conclusions in a separate paragraph.

• The following paragraph summarizing the results of the discussion in the form of conclusions has been added to Pages 14-15, Lines 322-332. 

“In conclusion, serum levels of C-reactive protein statistically correlate with and predict femoral neck and spine BMD in a sexually and racially diverse population. However, as the magnitude is too low to be biologically meaningful, significant caution should be taken in interpreting the clinical significance and application of serum C-reactive protein as a biomarker for osteoporosis. Weight more robustly predicts femoral neck and spine BMD, compared to serum C-reactive protein. Given that overweight and obese individuals account for nearly half of all osteoporotic fractures, the predictive power of weight in identifying individuals at risk for osteoporosis is severely limited. Current research should continue efforts to identify a clinically available biomarker that robustly predicts low bone mineral density and fracture risk in a diverse population and across of range of body weights and compositions. Rates of both osteoporosis and obesity continue to increase, with osteoporosis remaining largely undetected prior to first fracture, lending significant credence to the need for a robust and predictive biomarker.”

Comments relating to Journal Requirements:

1. The manuscript has been revised to ensure it meets PLOS ONE’s style requirements, including those for file naming.

2. I request no changes to the Funding Statement. I request the Acknowledgements Section be revised as follows, which has been updated in the Manuscript: 

“The author would like to thank Dr. Ayland Letsinger, Dr. Daniel Schmitt, and the Animal Locomotion Lab at Duke University for the much-appreciated conversations, feedback, and support.”

3. A full ethics statement has been included in the Methods section of the manuscript.

4. Captions for the Supporting Information materials have been included at the end of the manuscript according to PLOS ONE’s style guidelines. 

5. The Supporting Information file has been revised and re-uploaded. 

6. The reference list has been reviewed and is complete and correct.

---

## [Decision Letter · Decision Letter 1]

27 Sep 2023

Serum high sensitivity C-reactive protein poorly predicts bone mineral density: a NHANES 2017-2020 analysis

PONE-D-23-14590R1

Dear Dr. Sarah E Little-Letsinger,

We’re pleased to inform you that your manuscript has been judged scientifically suitable for publication and will be formally accepted for publication once it meets all outstanding technical requirements.

Kind regards,

Ewa Tomaszewska, DVM Ph.D

Academic Editor

PLOS ONE

Additional Editor Comments (optional):

Reviewers' comments:

Reviewer's Responses to Questions

**Comments to the Author**

1. If the authors have adequately addressed your comments raised in a previous round of review and you feel that this manuscript is now acceptable for publication, you may indicate that here to bypass the “Comments to the Author” section, enter your conflict of interest statement in the “Confidential to Editor” section, and submit your "Accept" recommendation.

Reviewer #1: All comments have been addressed

2. Is the manuscript technically sound, and do the data support the conclusions?

Reviewer #1: Yes

3. Has the statistical analysis been performed appropriately and rigorously? 

Reviewer #1: Yes

4. Have the authors made all data underlying the findings in their manuscript fully available?

Reviewer #1: Yes

5. Is the manuscript presented in an intelligible fashion and written in standard English?

Reviewer #1: Yes

6. Review Comments to the Author

Reviewer #1: I have re-reviewed the manuscript titled "Serum High Sensitivity C-Reactive Protein Poorly Predicts Bone Mineral Density: A NHANES 2017-2020 Analysis" submitted for publication in PLOS One as a research article.

The article has been supplemented in accordance with the suggestions from the first review.

I have no further objections.

7. PLOS authors have the option to publish the peer review history of their article (what does this mean?). If published, this will include your full peer review and any attached files.

Reviewer #1: No

---

## [Editor Report · Acceptance letter]

4 Oct 2023

PONE-D-23-14590R1 

Serum high sensitivity C-reactive protein poorly predicts bone mineral density: a NHANES 2017-2020 analysis 

Dear Dr. Little-Letsinger:

I'm pleased to inform you that your manuscript has been deemed suitable for publication in PLOS ONE. Congratulations! Your manuscript is now with our production department. 

Kind regards, 

on behalf of

Professor Ewa Tomaszewska 

Academic Editor

PLOS ONE